# Kinematic Alterations with Changes in Putting Distance and Slope Incline in Recreational Golfers

**DOI:** 10.3390/bioengineering12010069

**Published:** 2025-01-15

**Authors:** Shawn M. Robbins, Philippe Renaud, Ukadike Chris Ugbolue

**Affiliations:** 1School of Physical and Occupational Therapy, McGill University, Montreal, QC H3G 1Y5, Canada; 2Centre for Interdisciplinary Research in Rehabilitation and Lethbridge-Layton-MacKay Rehabilitation Centre, Montreal, QC H4B 1T3, Canada; 3Department of Kinesiology and Physical Education, McGill Research Centre for Physical Activity and Health, McGill University, Montreal, QC H2W 1S4, Canada; philjrenaud@hotmail.com; 4Biomechanics Laboratory, Division of Sport and Exercise, School of Health and Life Sciences, University of the West of Scotland, Paisley PA1 2BE, UK; u.ugbolue@uws.ac.uk; 5Faculty of Sports Science, Ningbo University, Ningbo 315211, China

**Keywords:** golf, putting, motion capture, kinematics, biomechanics

## Abstract

Golfers must modify their motor patterns when the demands of a putting task change. The objective was to compare joint angles and putter kinematics during putting at two distances and inclines. Recreational golfers (*n* = 14) completed putts over four conditions: 3-foot putts on flat and incline surfaces, and 7-foot putts on flat and incline surfaces. A Vicon motion capture system measured kinematic data. Joint angles, putter angles, and spatiotemporal variables were calculated. Analysis of variance compared spatiotemporal variables, and statistical parametric mapping compared angles between putts. There were faster putter head and ball velocities during longer and incline putts. The amplitude and time of backswing increased with longer putts. Longer putts resulted in increased trunk axial rotation during backswing, downswing, and follow-through, while incline putts only resulted in greater rotation during follow-through. There were minimal differences in shoulder angle. There was greater head rotation toward the hole during all putting phases for longer putts and during follow-through for incline putts. The trunk is the primary mechanism to increase putter head amplitude, and thereby velocity, when putting from longer distances. A similar strategy could be used when putting uphill. Additional work should confirm these results in highly skilled golfers.

## 1. Introduction

The game of golf is popular throughout the world, with an estimated 45 million golf participants in the United States [1] and 62.3 million in the rest of the world [2]. Golf presents a unique opportunity to study motor learning principles as the outcome of a golf stroke is simple (i.e., accuracy of the shot); it is played in a relatively controlled environment, and there are variations in the task demands [3]. One aspect to study is variation in motor patterns in response to changing task demands, including golfers modifying their body kinematics when putting from different distances or using different clubs. The success of motor learning training strategies (e.g., augmented feedback, cognitive training, explicit learning, and internal focus of attention) at improving outcomes and motor patterns can also be examined in golfers [3]. Considering the complexity of the golf swing, numerous studies have used biomechanical tools to evaluate and quantify the motor patterns or kinematics of this activity [4]. Most studies have examined the full golf swing, with golfers using drivers or wedges. Less frequently studied is the putting stroke.

On the Professional Golfers Association Tour in 2024, the average putts per round and total strokes per round were 29 and 71, respectively [5]. Thus, putting accounted for 40% of all strokes. For the current study, the putter refers to the club used in the putting stroke, and the putter head refers to the part of the club that contacts the ball. The putting stroke has been modelled as a double pendulum, which includes the putter and both arms [6,7]. To accomplish this motion, the “shoulder is meant to roll in an up-and-down fashion, and the two hands hold the putter together, moving back and forth to putt symmetrically” [8]. The putter face should remain perpendicular to the initial ball direction line [9,10]. The ball should also be hit on the upward stroke, which increases topspin, improving the efficiency of the ball roll and reducing skidding [10]. Head movement during putting is considered poor technique, and the golfer’s head should remain stationary [11].

Golfers putt from various distances, and adapting to different distances can be challenging. Typically, golfers should increase the amplitude of their stroke when putting from longer distances, while maintaining a consistent tempo [12,13]. In support, previous research has demonstrated increased amplitude of the putter head during the backswing, downswing, and follow-through phases of the putting stroke when putting from longer distances in highly skilled golfers [14,15,16]. The duration of backswing and downswing also increased with longer distances [14,15]. This increased backswing amplitude will naturally increase the putter head velocity, and faster putter head velocity has been demonstrated with longer putts [14,15,16,17]. Putter head velocity has been shown to be greater in people with no putting experience compared to expert golfers [14], although these difference were not present when comparing recreational to expert golfers [11].

Most studies have examined putter kinematics, and there has been limited research examining body kinematics during putting [11]. One study found that expert golfers had lower displacements of their heads compared to less skilled golfers during putting, although the results were not statistically significant [11]. This indicates that the head should remain stable when putting. In expert golfers, there were differences in thorax and lumbar spine angles between proficient (>79% putting success) and non-proficient (<79% putting success) golfers when completing 2 m putts [6]. Research is required to examine how kinematics of the golfer’s body change between putting distances. We know that putter kinematics change with increasing distances, but it is not clear how golfers produce these changes by modifying their joint angles. Furthermore, other task constraints have not been examined, including comparing putting between flat and inclined surfaces. Therefore, the objective was to compare joint angles, putter angles, and timing variables during putting at different distances and surface inclines in recreational golfers. It was hypothesized that there would be increased amplitudes in putter and shoulder angles, and faster ball and putter head velocities with longer distance and incline slopes.

## 2. Materials and Methods

### 2.1. Participants

This cross-sectional study was a secondary analysis of an existing database. The sample included 14 male recreational golfers. All golfers were right-handed, and all had experience playing golf. Exclusion criteria included current back, wrist, elbow, or arm injuries and additional injuries/diseases that would prevent them from playing golf. The study was approved by the School of Health and Life Sciences Ethics Committee at the University of the West of Scotland (#5-3-14-002). All participants provided written, informed consent.

Data were collected at the Biomechanics Laboratory at the University of the West of Scotland (Lanarkshire campus). Participants attended one data collection session. Demographic information was collected (e.g., age). The handicap and golfing experience (e.g., frequency of play, how long they had been playing) of the participants were not known.

### 2.2. Putting Task

Participants completed putts on an artificial mat (SKLZ Accelerator Pro Putting Mat, Carlsbad, CA, USA), which was attached to a wooden board (8 feet long, 1 foot wide). Participants used a standard putter (Classic Smoke Daytona 79 Putter, 34-inch shaft length, TaylorMade, Basingstoke, UK) and ball (Pro V1 golf ball, Titleist, Cambridgeshire, UK). During putting (Figure 1), participants stood on wooden blocks (30 × 10 cm) so they were level with the putting surface. Participants completed putts over four conditions: 3-foot flat, 7-foot flat, 3-foot incline, and 7-foot incline. To create the incline condition, extra wooden blocks were placed at the end of the putting surface and under the left foot of the participants to create a 5.7° incline. Participants were permitted a 5 min warm-up to become familiar with the equipment. They then completed three trials for each condition, and the order was randomized. They were instructed to putt the ball in the hole, and no additional instructions about technique were provided. Both successful and unsuccessful putts were included.

### 2.3. Biomechanical Data Collection

Kinematic data were recorded with an eight-camera motion capture system (Bonita cameras, Vicon Metrics Ltd., Oxford, UK) at 250 Hz. Vicon Nexus software (version 2.11, Vicon Metric Ltd., Oxford, UK) was used to record the data. Retroreflective markers (14 mm) were placed on the participants according to the Plug-in Gait marker system [18]. Four markers were placed on the putter, including three on the shaft (top, mid, and bottom) and one on the heel of the putter head. The golf ball was covered in retroreflective tape to ensure that it was visible during the data capture trial sessions. Additional 14 mm markers were placed on the spinal processes (C7 to L5) but were not used in the current study. In total, 55 markers were placed on the participant and putter. Relevant anthropometric measures were taken first, including height, weight, leg length, joint widths (knee, elbow, wrist, and hand), and shoulder offset. Participants then completed a static standing trial before completing the putting conditions.

### 2.4. Data Processing

Data were processed with Visual3d (version 2024.05.3, HAS Motion, Kingston, ON, Canada). Maker data were filtered with a lowpass, 4-order Butterworth filter with a 5 Hz cutoff [17]. The segments were defined according to the Plug-In Gait model [19]. Joint angles were calculated for the head (with respect to the trunk), trunk (with respect to the pelvis), and shoulder from the trail (right) side (arm with respect to the trunk) using Euler XYZ, YXZ, and ZYZ sequences, respectively, consistent with either previous golf studies (trunk) or ISB recommendations (shoulder) [20,21,22]. Data for the left shoulder were not consistently available and were not presented. Joint angles included in statistical analyses were head and trunk extension/flexion (positive = extension), sideflexion (positive = right sideflexion), and axial rotation (positive = left rotation). For the shoulder, plane of elevation (0° = horizontal abduction, 90° = forward flexion) and elevation (negative = elevation) were analyzed. These angles were considered since they would likely contribute to the putting stroke or could represent an error in putting (e.g., head rotation). Additionally, the putter angle relative to the lab coordinate system was determined using an Euler XYZ sequence. The rotation of the putter angle about the Y-axis was considered, with positive values representing putter head rotation toward the hole and negative values rotation away from the hole.

The putting stroke was divided into three phases: backswing, downswing, and follow-through [17,23]. Backswing began when the putter head marker velocity started moving in the negative direction (away from the hole) until the putter head marker reached its position furthest from the hole. Downswing then began and concluded when the ball reflective tape velocity started to increase, representing ball impact. Follow-through began at ball impact and concluded when the putter head marker velocity switched from a positive value (toward the hole) to a negative value. The time of each putting phase was determined, along with the peak ball and putter head velocity. Joint and putter angle waveforms were time-normalized to 100% of the putting stroke (backswing to follow-through). The three trials from each condition were then averaged for each participant.

### 2.5. Statistical Analysis

Since this was a secondary analysis of an existing database, a formal sample size calculation was not conducted. Descriptive statistics were calculated for the study variables. Two-way repeated analysis of variance (ANOVA) compared peak ball velocity, putter head velocity, and the times of each phase between the distance (3 vs. 7 feet) and incline (flat vs. incline) conditions. ANOVAs were conducted using SPSS (version 24, IBM Corp., Armonk, NY, USA). Statistical parametric mapping (SPM) two-way repeated measures ANOVAs compared joint and putter angle waveforms between distance and incline conditions [24]. The time node was each 1% of the putting stroke. A critical threshold was determined, and F-statistics that surpassed this critical threshold were deemed statistically significant (*p* < 0.05). The SPM analysis was completed with the spm1D toolbox in Matlab (version 2018a, MathWorks Inc., Natick, MA, USA) [25,26].

## 3. Results

### 3.1. Ball and Putter Head Velocity

There were statistically significant differences in peak ball velocity between distance (*p* < 0.001) and incline (*p* < 0.001) conditions, although the interaction was not significant (*p* = 0.337). Likewise, there were statistically significant differences in peak putter head velocity between distance (*p* < 0.001) and incline (*p* < 0.001) conditions, and the interaction was not significant (*p* = 0.200). There were faster peak ball and putter head velocities during the 7-foot and incline putts (Table 1).

### 3.2. Phase Times

There were significant differences in backswing times between distances (*p* = 0.001), but incline (*p* = 0.834) and interaction (*p* = 0.416) effects were not significant (Table 2). Backswing times were longer during 7-foot compared to 3-foot putts (mean difference = 0.06 s). There were no significant differences in downswing times for distance (*p* = 0.614), incline (*p* = 0.065), and interaction (*p* = 0.710) effects (Table 2). There were significant differences in follow-through times for the distance effect (*p* = 0.042), but incline (*p* = 0.736) and interaction (*p* = 0.737) effects were not significant (Table 2). Follow-through times were longer during 7-foot compared to 3-foot putts (mean difference = 0.03 s).

### 3.3. Putter and Joint Angles

Significant SPM distance and incline effects are discussed below. There were no significant interactions. There were greater putter angles away from the hole (negative values) during the second half of backswing and early downswing (28 to 58% putting stroke) and greater angles toward the hole (positive values) during follow-through (74 to 100%) for 7-foot compared to 3-foot putts (Figure 2). Additionally, there were greater angles toward the hole during follow-through (75 to 100%) for incline compared to flat putts. There was also greater trail-side shoulder elevation (more negative values) at the end of follow-through for 7-foot compared to 3-foot putts (Figure 2).

The 7-foot putts resulted in greater extension of the participants’ heads throughout the putt (0 to 100%) and greater rotation toward the hole (left rotation) during parts of backswing, downswing, and follow-through (26 to 69% and 79 to 100% putting stroke) (Figure 3) compared to 3-foot putts. Also, there was greater rotation of the participants’ heads toward the hole at the end of the backswing (94 to 100%) for incline compared to flat putts (Figure 3).

The 7-foot putts resulted in greater trunk sideflexion away from the hole (right sideflexion) during follow-through (69 to 100%) compared to 3-foot putts. There was greater axial trunk rotation away from the hole (right rotation) during backswing/downswing (25 to 59%) and toward the hole (left rotation) during follow-through (75 to 100%) for 7-foot compared to 3-foot putts (Figure 4).

## 4. Discussion

This study was novel since there is limited data evaluating changes in body kinematics when putting from different distances and incline conditions. The participants modified their movement patterns in response to the varying task demands. When putting from longer distances, recreational golfers increased the amplitude and time of their backswing, thereby increasing putter head velocity. This was likely accomplished by an increase in trunk axial rotation with longer putts, with minimal changes in shoulder angle. When putting on an incline, there was an increase in putter head velocity without an increase in putter head amplitude during backswing. These findings indicate that the trunk is a mechanism to increase putter head amplitude, and thereby velocity, when putting from longer distances. A similar approach should be considered when putting uphill. Results should be interpreted cautiously since participants were recreational golfers, and results cannot be generalized to professional golfers.

Consistent with our hypothesis, participants increased putter angle amplitudes during backswing, leading to longer backswing times, when putting from longer distances. This resulted in faster putter head and ball velocities. The follow-through putter angle amplitude and time also increased due to the faster putter head velocities. However, there was no change in the downswing times. In support, previous research demonstrated an increase in putter head amplitude and velocity, and in backswing times when putting from longer distances, in golfers with varying skill levels from novice to highly skilled [13,14,15,16,17,27]. Thus, coaches should encourage golfers to increase their putter angle amplitudes during backswing when golfers are adjusting to longer putts. For incline putts, there were faster putter head and ball velocities compared to the flat putts. Inconsistent with our hypothesis, there was no increase in the putter angle amplitude during backswing or the time of this phase. This finding likely indicates that participants used other mechanisms to increase their putter head velocity, such as swinging the putter harder, when putting on the incline surface. From a performance perspective, recreational golfers should also increase their putter angle amplitudes during backswing to have the necessary putter head velocity when putting uphill. The current findings might not be generalizable to highly skilled golfers as differences in putting kinematics have been found between skill levels [10,14,17,28]. Future research should evaluate how highly skilled golfers adapt to sloped surfaces, and if training intervention using motor control principles can help golfers of all skill levels adapt to varying putting distances and slopes.

To perform longer putts, participants increased their trunk axial rotation. This included greater axial rotation away from the hole (right rotation) during backswing and toward the hole during follow-through (left rotation) during longer putts (Figure 4). This motion would increase putter angle, thereby increasing putter head velocity, allowing them to putt further. Research investigating golf swings with drivers and irons also found that increased trunk axial rotation was related to clubhead speed [29]. For incline putts, there were small, nonsignificant increases in trunk axial rotation (Figure 4), and thus other mechanisms likely contributed to the faster putter head velocities during incline putts. We expected increased trail shoulder angle amplitudes with longer and incline putts. However, there was only a small increase in trail shoulder elevation during follow-through for longer putts, with differences generally being around 1°. Thus, increased putter angle amplitudes and faster putter head velocities during longer putts are more likely generated by trunk axial rotation. Longer distances also resulted in greater trunk right sideflexion (away from the hole) during follow-through, which may help maintain the centre of mass within the base of support as the putter swings in the opposite direction (toward the hole). Few studies have examined trunk and shoulder angles during putting, and thus it is difficult to locate our findings within the literature. One study found greater thorax and lumbar motions in the frontal plane (sideflexion) in proficient putters, while nonproficient putters had greater thorax and lumbar flexion [6]. Our study provides preliminary evidence that golfers should increase their axial trunk rotation to increase putter head velocity when putting from longer distances. However, more research is needed to examine how the body, and not just the putter, adapts to different putting conditions, especially in highly skilled golfers.

The position of the participants’ heads was also impacted by putting conditions. The heads of the participants were more extended throughout the putting stroke for longer distances. However, this was less than 2° generally, except differences were larger at the end of follow-through (Figure 3). There was greater left rotation (toward the hole) of the participants’ heads during parts of backswing and downswing, and at the end of follow-through for longer putts. This latter finding was also present when putting from an incline compared to a flat surface. These findings indicate that participants lifted their heads when putting from longer distances, which is considered poor putting technique [11]. Lifting the head near the end of follow-through is likely not a concern. Hower, head rotation during backswing and downswing could negatively impact longer distance putts by altering body position and, ultimately, putter path and should be discouraged in golfers. The current sample consisted of recreational golfers, and these findings may vary in professional golfers. In support, a previous study of expert golfers (*n* = 5) and less skilled golfers (*n* = 11) found that expert golfers had less head displacement during putting, although the results were not statistically significant [11].

This study had several limitations. The success of the putts was not recorded and thus relationships between body and putter kinematics with putting success could not be explored. All participants were recreational male golfers, and their handicaps and golfing experience (e.g., frequency) were not known. These golfers were included since their body kinematics were available in the database. Results cannot be generalized to highly skilled/professional golfers or female golfers. The sample size of the database was small and should be increased in future studies. There were not sufficient marker data to calculate left shoulder, bilateral elbow, or bilateral wrist angles. Their role in the putting stroke should be considered. Although putter and body kinematics are important when evaluating putting performance, other factors should also be considered in future studies, including the ability of golfers to read the green [9].

## 5. Conclusions

In conclusion, recreational golfers changed their motor patterns in response to different task demands. They increased the amplitude of their backswing, resulting in faster putter head velocities when putting from longer distances. Increases in trunk axial rotation were likely the primary mechanism to accomplish this greater backswing amplitude. When putting uphill, participants increased their putter head velocity without a significant increase in backswing amplitude or trunk axial rotation. Thus, they relied on other mechanisms to meet the required velocity demands of putting uphill. Future research should examine more skilled golfing populations (e.g., professionals) and female golfers, examine kinematics in more joints, and explore relationships between body kinematics and putting success.

## Figures and Tables

**Figure 1 bioengineering-12-00069-f001:**
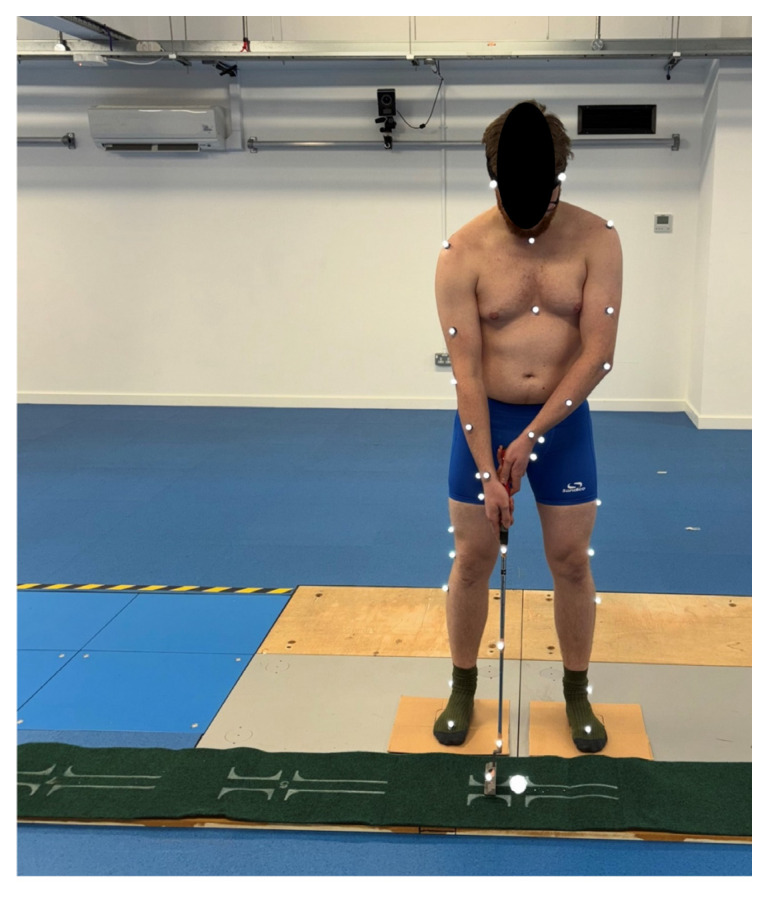
Data collection setup. During testing, we ensured the artificial putting surface was flat with no wrinkles.

**Figure 2 bioengineering-12-00069-f002:**
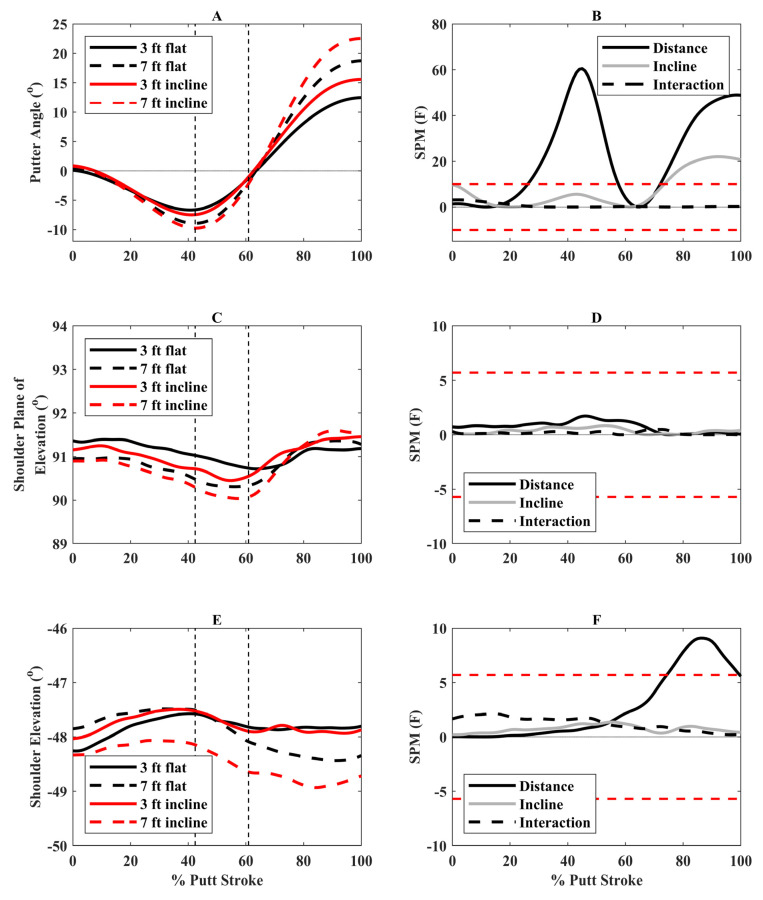
(**A**) Putter angles (positive = putter rotation toward hole), (**C**) Trail shoulder plane of elevation angles (0° = horizontal abduction, 90° = forward flexion), and (**E**) Trail shoulder elevation angles (negative = elevation) for the four putting conditions over the putting stroke (0° = start of backswing, 100° = end of follow-through). The vertical dashed lines represent the average transition between backswing and downswing (42.4% of putting stroke) and between downswing and follow-through (60.9% of putting stroke). Associated statistical parametric mapping (SPM) (**F**) plots are in panels (**B**,**D**,**F**). If the scalar statistic for each effect (distance, incline, interaction) goes above or below the critical threshold (red dashed lines), this indicates significant differences between putting conditions for that specific effect.

**Figure 3 bioengineering-12-00069-f003:**
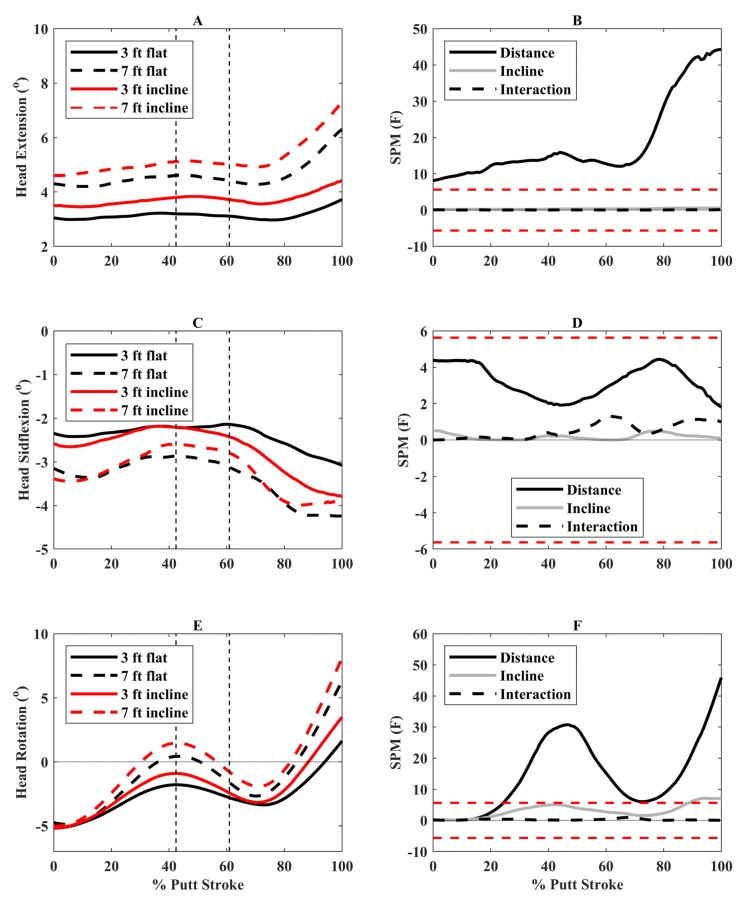
(**A**) Head extension angles (positive = extension), (**C**) Head sideflexion angles (positive = right sideflexion), and (**E**) Head axial rotation angles (positive = left rotation) for the four putting conditions over the putting stroke (0° = start of backswing, 100° = end of follow-through). The vertical dashed lines represent the average transition between backswing and downswing (42.4% of putting stroke) and between downswing and follow-through (60.9% of putting stroke). Associated statistical parametric mapping (SPM) (**F**) plots are in panels (**B**,**D**,**F**). If the scalar statistic for each effect (distance, incline, interaction) goes above or below the critical threshold (red dashed lines), this indicates significant differences between putting conditions for that specific effect.

**Figure 4 bioengineering-12-00069-f004:**
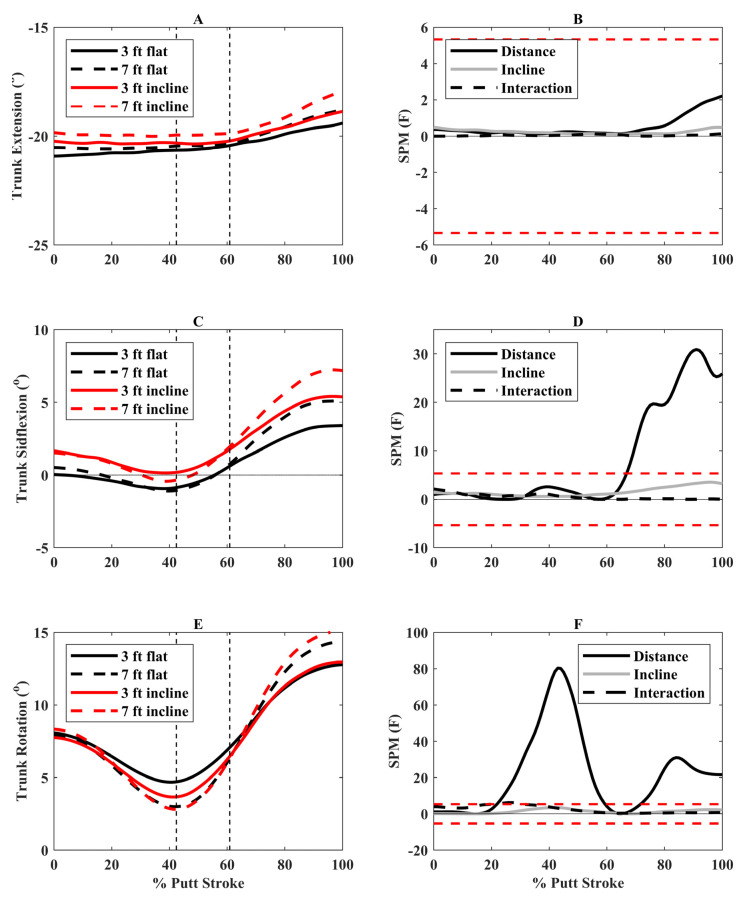
(**A**) Trunk extension angles (positive = extension), (**C**) Trunk sideflexion angles (positive = right sideflexion), and (**E**) Trunk axial rotation angles (positive = left rotation) for the four putting conditions over the putting stroke (0° = start of backswing, 100° = end of follow-through). The vertical dashed lines represent the average transition between backswing and downswing (42.4% of putting stroke) and between downswing and follow-through (60.9% of putting stroke). Associated statistical parametric mapping (SPM) (**F**) plots are in panels (**B**,**D**,**F**). If the scalar statistic for each effect (distance, incline, interaction) goes above or below the critical threshold (red dashed lines), this indicates significant differences between putting conditions for that specific effect.

**Table 1 bioengineering-12-00069-t001:** Demographic statistics and peak ball and putter head velocity for the study sample.

Variable		Mean (Standard Deviation)
Age (y)		22 (12)
Mass (kg)		77.86 (13.12)
Height (m)		1.78 (0.06)
Body mass index (kg/m^2^)		24.63 (3.42)
Peak ball velocity (m/s)	3-foot flat	1.29 (0.08)
	3-foot incline	1.68 (0.22)
	7-foot flat	2.08 (0.35)
	7-foot incline	2.57 (0.29)
Peak putter head velocity (m/s)	3-foot flat	0.93 (0.07)
	3-foot incline	1.16 (0.12)
	7-foot flat	1.33 (0.14)
	7-foot incline	1.62 (0.16)

**Table 2 bioengineering-12-00069-t002:** Mean (standard deviation) for each putting phase for the four conditions.

Condition	Backswing	Downswing	Follow-Through
3-foot flat (s)	0.53 (0.07)	0.25 (0.06)	0.52 (0.20)
3-foot incline (s)	0.53 (0.08)	0.24 (0.05)	0.51 (0.17)
7-foot flat (s)	0.58 (0.09)	0.25 (0.05)	0.54 (0.15)
7-foot incline (s)	0.59 (0.10)	0.24 (0.05)	0.55 (0.20)

## Data Availability

Data are not available due to privacy concerns.

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
