# Peer review of "Kinematic Alterations with Changes in Putting Distance and Slope Incline in Recreational Golfers"

_bioengineering, 2025, doi:10.3390/bioengineering12010069_

Round 1

Reviewer 1 Report

Comments and Suggestions for Authors

The proposed work investigates how joint angles and putter kinematics vary with putting distance and slope incline. Motion capture was employed to record the movements of golf players in varying conditions. The analysis of ball and putter velocities shows that longer putts and inclined surfaces result in higher velocities. The results show that body trunk motion is key in adjusting for distance and slope.

My understanding of the research question attempted to be answered, is whether and how the trunk motion of golfers affects the velocity and trajectory of a golf ball, in the context of the putting action.

The literature is appropriately reviewed and the paper reads in good English.

The motion capture procedure is appropriately conducted with a high-accuracy apparatus (VICON) and the measurements are reliable.

I am slightly surprised by the fact that all the subjects of the experiment are males. How can we know if females move in the same way? If not experimenting with females then the results cannot be claimed to hold for all recreational golfers but only for male recreational golfers. Is there a reason for this cohort selection? For example, in DOI:10.25904/1912/43 a similar study (for experienced golfers) includes both male and female participants.

Overall, the paper is correct and the analysis is appropriate. On the other hand, the results are not surprising as other studies also focus on trunk motion, e.g.  DOI:10.1080/02640414.2011.576693. Therefore if the value of this study is on inexperienced golf players, a relevant discussion would be helpful for the reader - particularly as to its motivation.

Author Response

We would like to thank the reviewers for their thoughtful comments. We have provided a response to each of their concerns below. Changes in the manuscript are highlighted. 

Comments 1: The literature is appropriately reviewed and the paper reads in good English.
Response 1: Thank you for the positive feedback.

Comments 2: The motion capture procedure is appropriately conducted with a high-accuracy apparatus (VICON) and the measurements are reliable.
Response 2: Thank you for the positive feedback.

Comments 3: I am slightly surprised by the fact that all the subjects of the experiment are males. How can we know if females move in the same way? If not experimenting with females then the results cannot be claimed to hold for all recreational golfers but only for male recreational golfers. Is there a reason for this cohort selection? For example, in DOI:10.25904/1912/43 a similar study (for experienced golfers) includes both male and female participants.
Response 3: We agree that not recruiting female golfers is a limitation. We used an existing database of male golfers, and thus data from female gofers were not available. We have added text to the discussion to clearly state this limitation (Line 312-315), and we mention that female golfers should be enrolled in future studies (Line 329).

Comments 4: Overall, the paper is correct and the analysis is appropriate. On the other hand, the results are not surprising as other studies also focus on trunk motion, e.g.  DOI:10.1080/02640414.2011.576693. Therefore if the value of this study is on inexperienced golf players, a relevant discussion would be helpful for the reader - particularly as to its motivation.
Response 4: We agree that other studies have measured trunk motion during the golf swing, but these studies explored swings with drivers or irons. Our research is novel since few studies have examined joint kinematics during putting, especially comparing kinematics when putting from different distances and slope conditions (Line 240-241). To address this comment, we added text and references (including DOI:10.1080/02640414.2011.576693) demonstrating the effect of trunk rotation on other golf swings (drives, iron play) (Lines 276-277). We also add a rationale in the limitations for why only recreational golfers were included (Line 313-314). We also state that future studies should explore joint kinematics during putting in highly skilled/professional golfers (Lines 269-271, Line 328-331).

Reviewer 2 Report

Comments and Suggestions for Authors

The manuscript titled "Kinematic Alterations with Changes in Putting Distance and Slope Incline in Recreational Golfers" presents an investigation into the biomechanics of golf putting. Putting, a crucial skill in golf, involves rolling the ball into the hole from short distances using relatively small and low-speed strokes.

Major Comments:

1. Abstract and Introduction:

The abstract begins with the objective: “The objective was to compare joint angles and putter kinematics during putting at two short distances and inclines.” While stating the objective is important, the authors should also consider adding a broader scientific context. Specifically, why is this study significant for the field of biomechanics or sports science? This will help contextualize the research and make it more relevant to readers. The context should be also added to the introduction as well.

In the discussion, the authors state, “This study was novel since there is limited data evaluating changes in body kinematics when putting from different distances and incline conditions.” However, the manuscript does not clearly explain how these novel findings can be applied or why they are important in the broader context of golf biomechanics or performance. A more explicit discussion of the potential implications of these findings for golfers, coaches, or future research would be valuable.

2. Figure 1:

Figure 1 depicts an artificial grass surface that appears uneven and wrinkled. Were any special measures taken to tighten or level the surface during the experiments? If so, this should be clarified in the manuscript. If not, the potential impact of the uneven surface on the results should be discussed, as it may affect the consistency of the putting stroke or introduce variability into the kinematic measurements.

3. Clarification of Terms (Line 188):

The term “head” in the sentence beginning with “The 7 foot putts resulted in …” is ambiguous. It is unclear whether the authors are referring to the “putter head” or the head of the player. To avoid confusion, the authors should clarify this distinction, as it could be misinterpreted by readers unfamiliar with the sport.

Additionally, it may be helpful to provide brief definitions of terms such as “putter” and “putter head” for clarity, particularly for readers who may not have prior knowledge of golf terminology. For instance, the word “putter” might be misread as referring to a player (i.e., the participant in the study), and the “putter head” could then be assumed to mean the participant’s head.

Minor points:

Line 42: the phrase “The putter face should remain square to the initial ball direction line” could be made clearer by replacing “square” with “perpendicular,” as “square” can be ambiguous.

Author Response

We would like to thank the reviewers for their thoughtful comments. We have provided a response to each of their concerns below. Changes in the manuscript are highlighted. 

Comments 1: The abstract begins with the objective: “The objective was to compare joint angles and putter kinematics during putting at two short distances and inclines.” While stating the objective is important, the authors should also consider adding a broader scientific context. Specifically, why is this study significant for the field of biomechanics or sports science? This will help contextualize the research and make it more relevant to readers. The context should be also added to the introduction as well.
Response 1: We recognize that this research could be placed in a broader context. We added additional text to the introduction (Line 32-40) and abstract (Line 13) stating how motor learning principles can be applied to golf. We had to cut some of the other text in the abstract to accommodate this latter change. We also added text to the discussion to ensure this theme is carried throughout the paper (Line 241-242).

Comments 2: In the discussion, the authors state, “This study was novel since there is limited data evaluating changes in body kinematics when putting from different distances and incline conditions.” However, the manuscript does not clearly explain how these novel findings can be applied or why they are important in the broader context of golf biomechanics or performance. A more explicit discussion of the potential implications of these findings for golfers, coaches, or future research would be valuable.
Response 2: In response to this comment, we have added additional text stating the implications of these findings for golfers and coaches (Line 259-260, Line 265-267, Line 305). We also added text stating potential avenues for future research (Line 269-271, Line 292-294).

Comments 3: Figure 1 depicts an artificial grass surface that appears uneven and wrinkled. Were any special measures taken to tighten or level the surface during the experiments? If so, this should be clarified in the manuscript. If not, the potential impact of the uneven surface on the results should be discussed, as it may affect the consistency of the putting stroke or introduce variability into the kinematic measurements.
Response 3: During testing, we ensured the artificial putting surface was tight with no wrinkles. The image was taken recently, after testing was completed, to ensure we had consent from the person in the image. We added the following sentence to the figure caption, “During testing, we ensured the artificial putting surface was flat with no wrinkles.”

Comments 4: The term “head” in the sentence beginning with “The 7 foot putts resulted in …” is ambiguous. It is unclear whether the authors are referring to the “putter head” or the head of the player. To avoid confusion, the authors should clarify this distinction, as it could be misinterpreted by readers unfamiliar with the sport.
Response 4: We recognize that there could be confusion. Throughout the paper, we refer to the head of the putter as “putter head”.  We have added text when referring to the head of the participants to provide greater clarity between the head of the participants and head of the putter (Line 193-197, Line 295-299).

Comments 5: Additionally, it may be helpful to provide brief definitions of terms such as “putter” and “putter head” for clarity, particularly for readers who may not have prior knowledge of golf terminology. For instance, the word “putter” might be misread as referring to a player (i.e., the participant in the study), and the “putter head” could then be assumed to mean the participant’s head.
Response 5: In response to this concern, we have added a sentence in introduction defining the terms “putter” and “putter head” (Line 46-47).

Comments 6: Line 42: the phrase “The putter face should remain square to the initial ball direction line” could be made clearer by replacing “square” with “perpendicular,” as “square” can be ambiguous.
Response 6: We have made this change as suggested (Line 51). 

Reviewer 3 Report

Comments and Suggestions for Authors

The paper presents a study on the body kinematics of amateur golfers during putting at two different distances and inclinations. While the study has significant limitations related to the sample size (few participants, all of whom are recreational golfers), which constrain its significance and generalizability, it is well-designed and executed, adhering to current quality standards. Additionally, the manuscript is well-structured and clear.

It is recommended that the authors strengthen the discussion section by emphasizing the potential practical utility of the study.

Author Response

We would like to thank the reviewers for their thoughtful comments. We have provided a response to each of their concerns below. Changes in the manuscript are highlighted. 

Comments 1: The paper presents a study on the body kinematics of amateur golfers during putting at two different distances and inclinations. While the study has significant limitations related to the sample size (few participants, all of whom are recreational golfers), which constrain its significance and generalizability, it is well-designed and executed, adhering to current quality standards. Additionally, the manuscript is well-structured and clear.
Response 1: Thank you for the positive comments.  We have added/revised comments in the limitations stating the limitation of the sample size and type of golfers (Line 312-316).

Comments 2: It is recommended that the authors strengthen the discussion section by emphasizing the potential practical utility of the study.
Response 2: In response to this comment, we have added additional text stating the implications of these findings for golfers and coaches (Line 259-260, Line 265-267, Line 305). We also added text stating potential avenues for future research (Line 269-271, Line 292-294).

Reviewer 4 Report

Comments and Suggestions for Authors

General comments

This manuscript aims to compare joint angles, putter angles, and timing variables during putting at different distances and surface inclines in recreational golfers. The authors’ aim is commendable. Authors found that there are faster putter head and ball velocities during longer and incline putts. Furthermore, the amplitude and time of backswing increase with longer putts, but do not change between incline and flat putts. Moreover, longer putts increase trunk axial rotation during the backswing, downswing and follow-through, while incline putts only result in greater rotation during the follow-through. In addition, there are minimal differences in shoulder angles between conditions. Finally, there is greater head rotation towards the hole during all putting phases for longer putts and during follow-through for incline putts. Overall, the authors fulfil their aim properly.

Specific comment

(lines 76-7) Please, quantify experience playing golf.

Minor comments

(line 195) Please, do not use acronyms in headings;

(l384) there is no 27 ref.

Author Response

We would like to thank the reviewers for their thoughtful comments. We have provided a response to each of their concerns below. Changes in the manuscript are highlighted. 

Comments 1: (lines 76-7) Please, quantify experience playing golf.
Response 1: The golfing experience of the participants was not known. We recognize that this is a limitation. We have added text in the methods (Line 92) and limitations (Line 312-313) to this effect. 

Comments 2: (line 195) Please, do not use acronyms in headings;
Response 2: In the submission we have, line 195 does not refer to a heading.  We checked the headings and subheadings, and there are no acronyms.  Were you referring to a heading in a figure or table?  We would be happy to make this change if you provide a reference to the subheading number or table/figure number. 

Comments 3: (l384) there is no 27 ref.
Response 3: The “27” was accidentally added by the editor.  It has been deleted.